# Drying-Time Study in Graphene Oxide

**DOI:** 10.3390/nano11041035

**Published:** 2021-04-19

**Authors:** Talia Tene, Marco Guevara, Andrea Valarezo, Orlando Salguero, Fabian Arias Arias, Melvin Arias, Andrea Scarcello, Lorenzo S. Caputi, Cristian Vacacela Gomez

**Affiliations:** 1Grupo de Fisicoquímica de Materiales, Universidad Técnica Particular de Loja, Loja EC-110160, Ecuador; tbtene@utpl.edu.ec; 2CompNano, School of Physical Sciences and Nanotechnology, Yachay Tech University, Urcuquí EC-100119, Ecuador; marco.guevara@espoch.edu.ec; 3UNICARIBE Research Center, University of Calabria, I-87036 Rende, Italy; andrea.valarezo@yachaytech.edu.ec (A.V.); orlando.salguero@yachaytech.edu.ec (O.S.); melvin.arias@intec.edu.do (M.A.); andrea.scarcello@unical.it (A.S.); lorenzo.caputi@fis.unical.it (L.S.C.); 4Facultad de Ciencias, Escuela Superior Politécnica de Chimborazo, Riobamba EC-060155, Ecuador; fabian.arias@espoch.edu.ec; 5Laboratorio de Nanotecnología, Area de Ciencias Básicas y Ambientales, Instituto Tecnológico de Santo Do-mingo, Av. Los Próceres, Santo Domingo 10602, Dominican Republic; 6Surface Nanoscience Group, Department of Physics, University of Calabria, Via P. Bucci, Cubo 33C, I-87036 Rende, Italy; 7INFN, Sezione LNF, Gruppo Collegato di Cosenza, Via P. Bucci, I-87036 Rende, Italy

**Keywords:** graphite, few-layer graphene, graphene oxide, Raman, TEM, UV-vis, Lorentzian fitting

## Abstract

Graphene oxide (GO) exhibits different properties from those found in free-standing graphene, which mainly depend on the type of defects induced by the preparation method and post-processing. Although defects in graphene oxide are widely studied, we report the effect of drying time in GO and how this modifies the presence or absence of edge-, basal-, and sp^3^-type defects. The effect of drying time is evaluated by Raman spectroscopy, UV-visible spectroscopy, and transmission electron microscopy (TEM). The traditional D, G, and 2D peaks are observed together with other less intense peaks called the D’, D*, D**, D+G, and G+D. Remarkably, the D* peak is activated/deactivated as a direct consequence of drying time. Furthermore, the broad region of the 2D peak is discussed as a function of its deconvoluted 2D_1A_, 2D_2A_, and D+G bands. The main peak in UV-visible absorption spectra undergoes a redshift as drying time increases. Finally, TEM measurements demonstrate the stacking of exfoliated GO sheets as the intercalated (water) molecules are removed.

## 1. Introduction

Graphene, a two-dimensional carbon nanomaterial arranged in hexagonal symmetry, has already demonstrated excellent electronic, mechanical, electric, magnetic, and thermal properties [1,2,3] which guarantee exciting applications, including composites [4], energy storage [5], catalysis [6], field-effect transistors [7], and plasmonics [8,9,10]. Several strategies are currently used for preparing graphene, for instance, chemical vapor deposition (CVD) [11], epitaxial growth [12], liquid-phase exfoliation [13], shear-exfoliation [14], zeolite-shear exfoliation [15], and chemical exfoliation [16]. 

Among them, the liquid-phase exfoliation and chemical exfoliation strategies are the most practical methods for preparing graphene in large quantities. In particular, the chemical exfoliation uses strong oxidizing agents to produce graphite oxide [17,18], which under sonication yields graphene oxide (GO) [19]. After the oxidation process, GO is covered by different functional groups (hydroxyl, epoxide, carboxyl, and carbonyl groups [20,21]) which increase the interlayer spacing up to 0.87 nm [22]. Moreover, GO is a large-band gap material [23], limiting its use for electronic applications, but opening a multitude of other applications as the removal of heavy metals [24] or dyes [25] as well as GO-based hydrogels [26]. 

Depending on the oxidation process, e.g., Hummers [27], Marcano [28], and Chen [29], GO could present different structural defects, for example, edge-, vacancy-, and sp^3^-type defects [30,31]. The study and control of such defects is of vital importance before choosing any of its widespread intended applications. In this respect, Raman spectroscopy is the most used characterization technique to scrutinize the quality of the as-made GO. The Raman spectra of graphite, single-layer graphene, few-layer graphene (FLG), and GO are widely reported in the literature, characterized by three prominent peaks, namely the D band, the G band, and the 2D band [32]. Less intense peaks called the D’, D*, D**, D+G, and G+D can also be observed [33]. 

The study of these bands is fundamental in characterizing graphene-derived materials. In this context, Kaniyoor et al. showed a study of the Raman spectrum of GO, considering seven different preparation strategies [33]. It is worth noting that the Raman spectrum of GO is significantly different from that of single-layer graphene (SLG) and must be carefully analyzed. Consequently, there is a lot of discrepancy in the literature over the Raman spectra of oxidized graphenes. Some reports show similarly intense D and G peaks with a highly broadened and low intense 2D band region [34,35,36]. In fact, in many published papers, the 2D band is neither shown nor discussed. 

The different strategies for preparing GO present environmental issues during the preparation process. As an example, Hummers et al. [27] reported the strategy most used: The oxidation process is carried out by treating natural graphite with KMnO_4_ and NaNO_3_ in concentrated H_2_SO_4_. This procedure involves the generation of toxic gases, such as NO_2_ and N_2_O_4_, limiting the large-scale production of GO. Recently, we have demonstrated a scalable eco-friendly protocol by excluding NaNO_3_ from chemical reaction [29,37] and subjecting the resulting graphite oxide to simple purification steps to obtain GO. Therefore, an investigation of the Raman spectrum of the as-made GO is not reported yet.

In this paper, such a study is presented. Instead of giving a comparison between GO preparation methods (oxidation or reduction strategies) as reported in [33], we report the effect of drying time to obtain GO powder, by using Raman spectroscopy. This study is complemented by transmission electron microscopy (TEM) and UV-visible (UV-vis) absorption measurements. For comparison purpose, the Raman spectra of natural graphite and FLG prepared in ethanol, are also reported.

## 2. Materials and Method

All chemicals were used as received, without further purification. Graphite powder (<150 μm, 99.99%), sulfuric acid (H_2_SO_4_, ACS reagent, 95.0–98.0%), potassium permanganate (KMnO_4_, ACS reagent, ≥99.0%), hydrochloric acid (HCl, ACS reagent, 37%), hydrogen peroxide (H_2_O_2_, 30%, Merk, Kenilworth, NJ, USA), and ethanol (purity ≥ 98.0%, CAS: 64–17–5) were obtained from Sigma-Aldrich (St. Louis, MO, USA). 

### 2.1. Synthesis of GO

GO was prepared as reported in our earlier paper [37]. Briefly, 3.0 g of graphite powder was added to 70 mL H_2_SO_4_ while stirring in an ice-water bath. Then, 9.0 g KMnO_4_ were added. The resulting mixture was transferred to an oil bath and agitated for about 0.5 h. After that, 150 mL distilled water was added, and the solution was stirred for 20 min. Additionally, 500 mL distilled water was added, followed by 15 mL H_2_O_2_ and stirred up to see a yellowish solution. The resulting graphite oxide suspension was washed with 1:10 HCl solution and distilled water eight times through centrifugation. The precipitated material was re-dispersed in water by sonication using an ultrasonic bath (Branson 2510 Ultrasonic Cleaner). The suspension was centrifugated at 1000 rpm for 0.5 h, and then, dried in a drying stove (2005142, 60 Hz, 1600 W, J.P. Selecta, Barcelona, Spain) at 80 °C for 0.5 h, 1 h, 3 h, 5 h, 24 h, taking 15 mL samples from the suspension. The obtained GO powder was used in subsequent characterization. 

### 2.2. Synthesis of Few-Layer Graphene

To perform the sonication process as simple as possible, 100 mg of graphite was added into 100 mL of ethanol using closed tube containers. The resulting mixture was sonicated employing an ultrasonic bath (Branson 2510 Ultrasonic Cleaner, 40 kHz, 130 W) in continuous operation. The sonication time was set to 7 h, and the resulting dispersion was centrifuged for 10 min at 1000 rpm to remove non-exfoliated graphite particles. Solvent evaporation is avoided because the sonication is made in sealed containers, and the temperature of the bath is controlled by fluxing fresh water every 0.5 h. 

### 2.3. Characterization

Raman spectra of graphite, FLG, and oxidized graphenes were obtained using a Jasco NRS-500 spectrometer with a 532 nm laser wavelength (0.3 mW, 100× objective). The surface morphologies of the samples were taken out on a transmission electron microscope (TEM, JEM 1400 Plus) operating at 80 kV. For TEM and Raman characterization, GO and FLG samples was prepared by drop casting onto formvar-coated copper grids, and glass substrates, respectively. Similarly, the GO samples subject to drying-time experiment were directly deposited on the corresponding substrates. The UV-visible measurements were recorded using a UV-vis spectrometer (Thermo Scientific, Evolution 220, Waltham, MA, USA), re-dispersing the treated GO by mid-sonication for 5 min.

## 3. Results and Discussion

Let us stress again, the main goal of the present work is the Raman study of the effect of the drying time on graphene oxide samples. As commented, [33] showed a detailed study of the Raman spectra of GO considering different preparations processes, and the effect of the temperature was widely understood [38]. The dependency of graphene oxide layers on drying methods was also briefly reported [39]. Very recently, the study of the effect of the Raman excitation laser on GO was reported [40].

### 3.1. Raman Spectrum of Graphite

Natural graphite has a crystal structure made up of flat graphene layers. These layers are stacked in a hexagonal honeycomb-like network, usually following the AB Bernal stacking or AA’ stacking. The interatomic in-plane distance is 1.42 Å, while the out-of-plane distance between graphene layers (due to van der Waals interactions) can have values ranging from is 3.35 Å up to 3.70 Å. 

The Raman spectrum of natural graphite is shown in Figure 1. The main feature is the first order spectrum displaying the E_2g_ in-plane optical mode (commonly called G peak) at 1577 cm^−1^ (Figure 1a). This narrow G peak appears due to the bond stretching of all pairs of sp^2^ hybridized carbon atoms in both rings and chains [41]. The G* peak found at 2447 cm^−1^ is characteristic of graphitic materials. The 2D peak appears at 2720 cm^−1^, characterized by two bands (Figure 1b), the intense 2D_2A_ band at 2720 cm^−1^, and a less intense 2D_1A_ band at 2677 cm^−1^. These bands are originated due to the splitting of *π* electrons as an effect of the interaction between stacked graphene layers.

The position and shape of the 2D peak depends, mainly, on the number of layers. Indeed, the 2D peak in SLG is fitted by a single Lorentzian function, say, the 2D_1A_ band which is located around 2680 cm^−1^ [42]. Moreover, the intensity ratio I_2D_/I_G_ in SLG is >11 while our starting graphite shows a I_2D2A_/I_G_ ≈ 0.45, typical of any natural graphite.

Hence, the G peak is related to the C−C stretching mode in sp^2^ carbon bonds and the 2D peak is a fingerprint to evaluate “qualitatively” the number of layers in the obtained graphene or graphene-derived material. The absent (or negligible intensity) of the D peak evidences a defect-free pristine graphite due to the D peak is ascribed to the basal/edge structural imperfections, corresponding with an increase in the amount of disorderly carbon and a decrease in the graphite crystal size [43].

### 3.2. Raman Spectrum of Few-Layer Graphene

The Raman spectrum of FLG prepared in ethanol by sonication was discussed in detail in Refs. [44,45]. The three significant peaks of FLG are depicted in Figure 2, the D peak at 1339 cm^−1^, the G peak at 1578 cm^−1^, and the 2D peak at 2719 cm^−1^. Additionally, other less intense peaks are detected, the D** peak at 1385 cm^−1^, the D’ peak at 1615 cm^−1^, the D+G at 2901 cm^−1^, and the G+D’ at 2961 cm^−1^.

Here, the D and D’ peaks are activated due to (i) the induced structural damage after the sonication process and (ii) the reduction of the lateral size of the graphene sheets. However, the D peak is narrow and significantly less intense than the G peak, suggesting edge defects (folded edge samples) rather than basal defects (vacancies or impurities). The intensity ratio I_D_/I_G_ was found to be <0.5. The type of defects can be deduced analyzing the D’ peak and the intensity ratio I_D_/I_D’_. In particular, the value of I_D_/I_D’_
~3.7 is associated to edge-type defects, but basal defects cannot be ruled out completely since the D** and D+G peaks are also present. Particularly, these peaks (with an appreciable intensity) have been found in highly disordered carbon samples. 

Therefore, the origin of these low-intensity peaks (D**, and D+G) in FLG appear as an effect of the structural damage due to the sonication time, solvent type, and sonication power, crucial parameters to be controlled to attain the scalable production of graphene dispersions. On the other hand, defects do not activate the G+D’ peak because of momentum conservation restrictions [46].

Compared to graphite, the 2D peak in FLG is also characterized by two bands, the 2D_1A_ at 2685 cm^−1^ and the 2D_2A_ at 2720 cm^−1^. Three main characteristics are observed: (i) The complete shape of the 2D peak changes, (ii) the intensity of the 2D_1A_ band increases close to that of the 2D_2A_ band, and (iii) the intensity ratio of I_2D2A_/I_2D1A_ is 1.12 (in graphite, I_2D2A_/I_2D1A_ = 2.78). All these evidences support the transformation of graphite into FLG, likely no more than 10 layers. 

### 3.3. Raman Spectrum of Non-Dried Graphene Oxide

We now move to the focus of this communication. The Raman spectrum of GO (obviously different from that of graphite and FLG) is shown in Figure 3, and the corresponding peak position and full-width at half maximum (FWHM) are reported in Table 1 and Table 2. Notice that the G+D’ peak is not detected in the wavenumber window analyzed from 1000 cm^−1^ to 3000 cm^−1^ (Figure 3b). This peak is assumed to be shifted at higher wavenumber values (>3100 cm^−1^) because of the spectral weight of the D+G peak.

It is well known that in GO, the D and D’ peaks are related to the presence of defects such as folded edges, vacancies, impurities (functional groups or remaining metal species), and the change from sp^2^ to sp^3^ hybridization [47,48,49,50]. In particular, a decrease in the D’ peak intensity can be considered as straight evidence of GO reduction. On the other hand, the intensity ratio I_D_/I_D’_ in our as-made GO is ~2.2, erroneously suggesting only the predominance of edge defects in its structure, even a lower value than that observed in FLG. 

The D** band is due to contributions from the phonon density of states in finite size graphitic crystals, C−H vibrations in hydrogenated carbon, and hopping-like defects [51]. Depending on the preparation process, GO also displays a D* band related to the sp^3^ diamond line on disordered amorphous carbons, but the broad region between ~1400 cm^−1^ and ~1650 cm^−1^ is not attributed to diamond carbon phases [31,52,53]. Although the D* band is not perceptible in non-dried GO, this band appears after the drying-time experiment (discussed below). 

Interestingly enough, the 2D band region (Figure 3b) is characterized by intense 2D_1A_ and D+G bands and a less intense 2D_2A_ band in contrast to those observed in natural graphite (Figure 1b) and FLG (Figure 2b). The intensity ratio I_2D2A_/I_2D1A_ = 0.22 in GO decreases about 12 times and 5 times compared to graphite and FLG, respectively. These outcomes are probably due to a good chemical exfoliation and reduction of the number of layers, but the resulting material has sufficient defects to activate the D, D’, D**, and D+G bands. 

### 3.4. Raman Spectrum of Dried Graphene Oxide 

Figure 4 shows the Raman spectra of GO dried at different times from 0.5 h to 24 h, keeping the temperature fixed (80 °C). The D, G, and D’ peaks are observed which are not substantially affected in position, FWHM, or intensity in the drying-time testing (Table 1, Figure 5). In fact, a thermal reduction of the material obtained is not expected, as confirmed by FT-IR results (no shown here). Interestingly, the D* band is not as intense at 0.5 h and 1 h, and can be seen as the drying time increases from 3 h to 24 h.

Two intervals are observed: (i) From 0.5 h to 3 h, the D* peak is shifted at 1135 cm^−1^ while the FWHM value remains relatively unchanged, and (ii) from 5 h to 24 h, the peak position is shifted to lower wavenumber values while the FWHM value notably increases, giving the highest value at 5 h (FWHM = 174) and 24 h (FWHM = 89). In contrast, the peak position and FWHM value of the D** peak are not clearly affected by the drying time, but its intensity increases along with the intensity of the D* band.

As seen in Figure 5, the relative intensity of the D’ (blue line) and G (orange line) bands shows a constant trend, while the D* (green line) and D** (magenta line) display a square-root-like dispersion. From the curve fitting, it can be seen that the most significant effect of drying time is ≤10 h. These outcomes show that it is not possible to transform GO into reduce graphene oxide (rGO) through drying time at 80 °C. Nevertheless, if it is possible to increase the density of sp^3^-type and hopping-like defects associated with the presence of the D* and D** band, respectively.

Therefore, it cannot be said that there is only one type of defects in GO. Instead of doing an analysis based on the intensity ratio I_D_/I_D’_, we hypothesize that the study of defects in GO must be accompanied through the ratio of intensities, i.e., I_D_/I_D’_ for edge-, I_D_/I_D*_ for sp^3^-, and I_D_/I_D**_ for hopping-like defects. In the present work, the corresponding intensity ratio values as function of the drying time are: 23.95 < I_D_/I_D*_
< 6.05, 6.14 < I_D_/I_D**_
< 3.31, and 1.70 < I_D_/I_D’_
< 1.76. This hypothesis and intensity ratio values should be corroborated future studies. Most importantly, it can be noted that the longer the drying time, the lower the intensity ratio of I_D_/I_D*_ and I_D_/I_D**_ while the intensity ratio I_D_/I_D’_ slightly increases. The I_D_/I_D’_ values reasonably agree with those reported in [54]. 

The presence of defects in GO also affects the 2D band region which is deconvoluted in the 2D_1A_, 2D_2A_, and D+G bands (Figure 6). In the literature, these bands are not discussed in detail when talking about conventionally prepared GO because the very low intensity and only noticeable when rGO is obtained, a sign of the recovery of the graphene structure. An important feature of GO is the presence of the 2D band region, suggesting non-critical damage to graphene structure after oxidation process. Thus, non-aggressive, environmentally friendly, and highly efficient reducing agents could be used, for instance, citric acid, ascorbic acid, or citrus hystrix [55].

The Raman spectra, peak position, FWHM, and relative intensity of the deconvoluted 2D band region as a function of drying time, are reported in Figure 6, Table 2 and Figure 7, respectively. The 2D_1A_ and D+G bands are clearly observed, while the 2D_2A_ band begins to appear as an effect of the drying time after 3 h, reaching an intensity comparable to that of the 2D_1A_ and D+G bands at 5 h (Figure 6d) and 24 h (Figure 6e). 

The decrease of the intensity of the 2D_2A_ band at 0.5 h (Figure 6a) and 1 h (Figure 6b), can be interpreted as due to GO with few layers and wrinkled structure, as can be concluded from the prevalence of the 2D_1A_ and D+G bands, respectively. The presence of the 2D_2A_ band after 3 h of drying is attributed to the evaporation of water molecules between the GO layers, causing the stacking of the chemically exfoliated sheets. Trying to explain this fact, we have carried out TEM measurements (discussed below).

Although the FWHM values of the 2D_1A_ band does not critically change after 1 h of drying, its position is shifted at 2685 cm^−1^. After that, the value of the FWHM decreases (e.g., FWHM = 188 at 24 h) and the peak position moves to lower wavenumber values (2653 cm^−1^ at 5 h and 24 h), even lower than that observed in graphite (2677 cm^−1^) or FLG (2685 cm^−1^). 

Compared to the peak position of the 2D_2A_ band in graphite or FLG (2720 cm^−1^), this band in GO is found at ~2750 cm^−1^ and is not affected by drying time, but the FWHM value decreases down to 46 cm^−1^, where the intensity of the 2D_2A_ band is barely perceptible (0.5 h and 1 h). On the other hand, the peak position and FWHM value of the D+G band decrease (e.g., 2922 cm^−1^ and FWHM = 131 cm^−1^ at 24 h) as the drying time increases. The peak position found at 5 h and 24 h of drying are close to that of FLG (2901 cm^−1^). The latter corroborates our statement that the D+G band moves to higher wavenumber values as an effect of the oxidation process, also causing the displacement of the G+D’ band. Therefore, it is demonstrated the D+G band has a high dependence on drying time, and its effect must be carefully considered when characterizing GO samples.

The intensity ratio I_2D2A_/I_2D1A_ increases from 0.22 to 0.89, a value close to that observed in FLG (I_2D2A_/I_2D1A_ = 1.12) while the intensity ratio I_D+G_/I_2D2A_ decreases from 1.21 to 1.14. The latter suggests a slight reduction of defects. Most importantly, Figure 7 shows the relative intensity of the 2D_1A_ (black line), 2D_2A_ (blue line), and D+G (red line) bands. The 2D_1A_ and D+G bands are characterized by an exponentially decreasing behavior while the 2D_2A_ band shows a sigmoid growth trend. From the curve fitting, it can be seen that the most significant effect is obtained for drying times ≤5 h. This crucial result shows that for long drying times, the water molecules and other possible oxygen-containing molecules are removed, allowing the exfoliated GO sheets to pile up, probably, to re-form graphite oxide.

With all this in mind, the Raman results showed that a GO not very disordered and with few layers can be obtained with a drying time of 1 to 3 h. After that, a highly disordered graphitic-like structure is expected, suggesting that long drying times being unnecessary in a practical large-scale GO production. Furthermore, long drying times may not facilitate the recovery of the graphene structure after even aggressive reduction processes, for example, using hydrazine. This statement motivates more extended works.

The UV-vis absorption spectra of GO subject to different drying times are depicted in Figure 8. At 0 h of drying, GO exhibits an absorption peak at ~233 nm and a shoulder at ~304 nm, which are attributed to the $\pi -\pi^{*}$ transition in C−C bonds and the $n-\pi^{*}$ transition in C=O bonds, respectively. As drying time increases, the peak and the shoulder gradually redshift at ~250 nm and at ~325 nm, respectively, suggesting that the electronic conjugation within graphene structure starts to be restored. Although it is not possible to affirm the transformation of GO into rGO (as evidenced by Raman measurements), this result offers the possibility to adapt the optical and electrical properties of GO.

In addition to the results obtained by the drying time experiment, we also briefly describe the effect of rehydration in Appendix A (Raman spectra), and Appendix A (UV-vis measurements). The dried GO samples (at 1 h, 5 h, and 24 h) were newly redispersed in continuous stirring for 30 min. Particularly, the D* band is not perceptible in all rehydrated samples, and the intensity of the 2D_2A_ band substantially decreases. UV-vis results evidence a blueshift of the absorption $\pi -\pi^{*}$ peak. These outcomes can be attributed a well redispersion of GO due to its hydrophilic properties.

### 3.5. TEM Analysis of Graphene Oxide

TEM micrographs of non-dried GO and GO subjected to different drying times from 0.5 h to 24 h are shown in Figure 9. Non-dried GO sample appears as a transparent and thin nanosheet with some wrinkles and folds on the surface and edges, but not with a very disordered surface morphology as GO prepared by conventional methods. Instead of point defects, these wrinkles are associated with surface defects formed due to the folding or twisting of the exfoliated GO sheets, causing deviation from the sp^2^ to sp^3^ character. This wrinkled structure is a characteristic of graphene-like materials, whereas they are not present in other carbon nanostructures, e.g., amorphous carbons. 

At a first approximation, the GO sample subject to 0.5 h of drying (Figure 9b) appears to be very similar to non-dried GO. However, a clear difference is observed beyond the wrinkles and folds, i.e., opaque regions are detected, which are attributed to the stacking of exfoliated GO sheets because the intercalated water molecules between the layers begin to be removed. This effect is more clearly observed after 1 h and 3 h of drying. At 5 h and 24 h, the GO samples look dark, suggesting a high removal of the intercalated molecules between layers, which causes a large stacking of the exfoliated GO sheets. This outcome supports the presence of the 2D_2A_ band observed in the Raman study. In particular, Figure 9f demonstrates the stacked layers after 24 h of drying, and a GO sample with a lateral size larger than the previous ones, which also allows to observe that the GO flakes are reassembling. The inset in Figure 9f shows that the flake with lateral size in the micrometers range is made of a superposition of restacked GO layers.

## 4. Conclusions

In summary, we have reported the effect of drying time (from 0.0 h to 24 h) on GO by a systematic Raman study. For comparison, the Raman spectrum of graphite and FLG were also discussed. The work is complemented using UV-vis and TEM measurements. The D, G, and D’ peaks were not affected by increasing the drying time, but the D*, D**, and 2D_2A_ peaks seemed to be very sensitive. The relative intensity of the different Raman bands and corresponding FWHM’s were discussed. UV-vis results evidenced a redshift from ~233 nm (0.0 h) to ~250 (24 h) nm. TEM results showed the stacking and reassembly of GO sheets as a direct consequence of the drying time. 

The study was carried out with a process as simple as possible, subjecting each sample to 80 °C in an oven, and characterizing them as soon as possible. For this reason, in future studies, a controlled environment should be considered as well as the time the samples remain at room temperature before being characterized. The water molecules would continue to evaporate for long periods. Additionally, we suggest to consider the type of graphite since here only graphite powder has been used. In fact, large lateral graphite or expanded graphite could present different results compared to those reported here. 

Our findings are intended to contribute to the control of the technical parameters involved in the synthesis process to achieve the large-scale production of GO powder.

## Figures and Tables

**Figure 1 nanomaterials-11-01035-f001:**
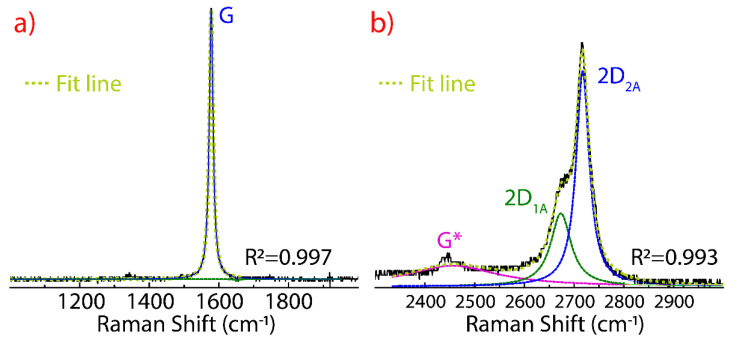
Raman spectrum of graphite (**a**) from 1000 to 2000 cm^−1^ and (**b**) from 2300 to 3000 cm^−1^ recorded using 532 excitation laser. The intensity was normalized by the most intense peak and the fitting of the peaks using Lorentzian functions.

**Figure 2 nanomaterials-11-01035-f002:**
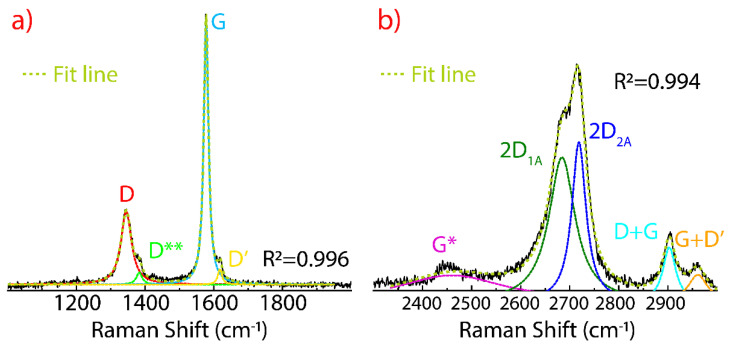
Raman spectrum of FLG (**a**) from 1000 to 2000 cm^−1^ and (**b**) from 2300 to 3000 cm^−1^ recorded using 532 excitation laser. The intensity was normalized by the most intense peak and the fitting of the peaks using Lorentzian functions.

**Figure 3 nanomaterials-11-01035-f003:**
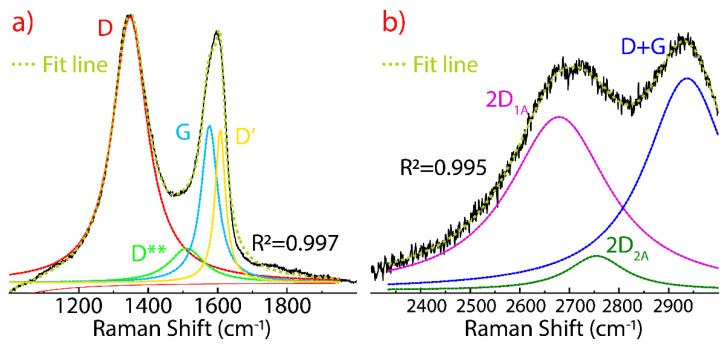
Raman spectra of GO (**a**) from 1000 to 2000 cm^−1^ and (**b**) from 2300 to 3000 cm^−1^ recorded using 532 excitation laser. The intensity was normalized by the most intense peak and the fitting of the peaks using Lorentzian functions.

**Figure 4 nanomaterials-11-01035-f004:**
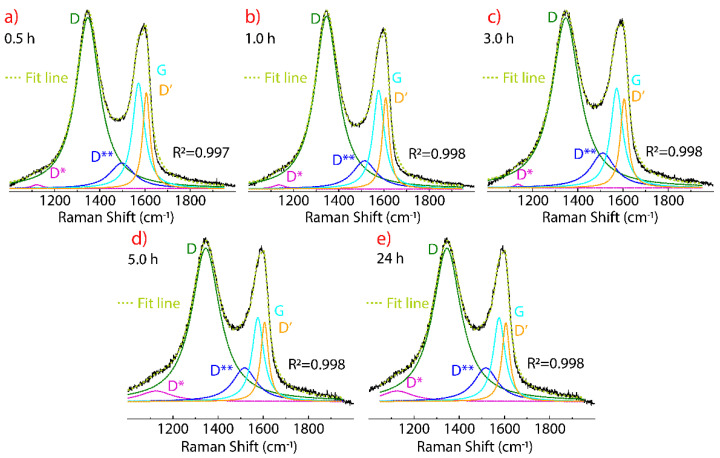
Raman spectra of GO from 1000 to 2000 cm^−1^ recorded using 532 excitation laser, subject to 80 °C, and considering different drying times: (**a**) 0.5 h, (**b**) 1 h, (**c**) 3 h, (**d**) 5 h, and (**e**) 24 h. The intensity was normalized by the most intense peak and the fitting of the peaks using Lorentzian functions.

**Figure 5 nanomaterials-11-01035-f005:**
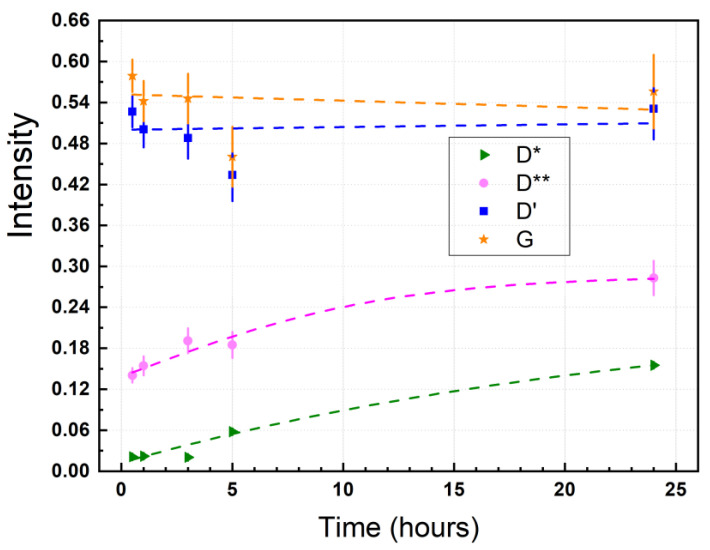
Intensity of the D*, D**, G, and D’ peaks as function of the drying time (0.5 h, 1 h, 3 h, 5 h, 24 h). The maximum intensity was obtained from the fitting using Lorentzian functions.

**Figure 6 nanomaterials-11-01035-f006:**
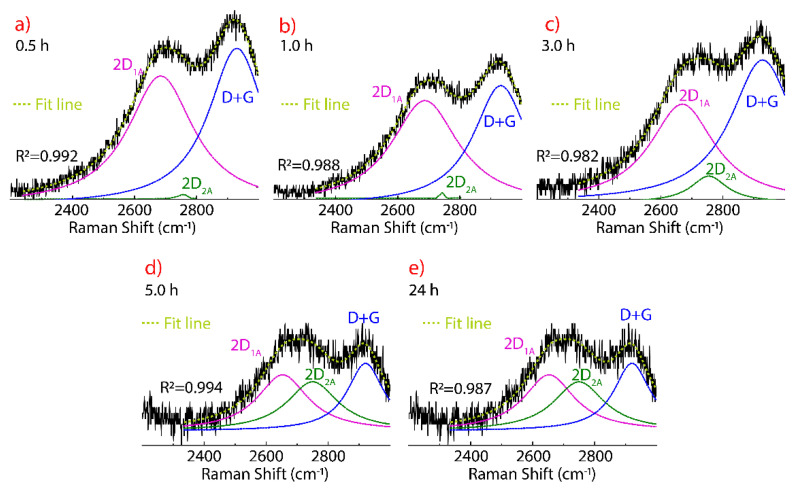
Raman spectra of GO from 2300 to 3000 cm^−1^ recorded using 532 excitation laser, subject to 80 °C, and considering different drying times: (**a**) 0.5 h, (**b**) 1 h, (**c**) 3 h, (**d**) 5 h, and (**e**) 24 h. The intensity was normalized by the most intense peak and the fitting of the peaks using Lorentzian functions.

**Figure 7 nanomaterials-11-01035-f007:**
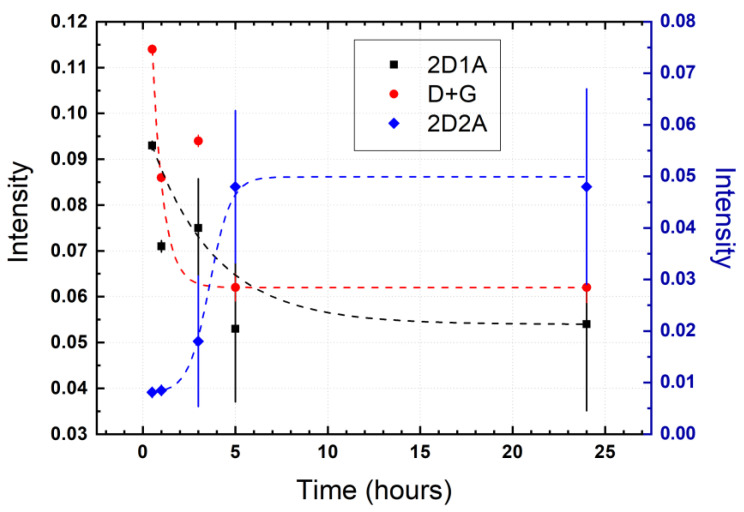
Intensity of the 2D_1A_, 2D_2A_, D+G peaks as function of the drying time (0.5 h, 1 h, 3 h, 5 h, 24 h). The maximum intensity was obtained from the fitting using Lorentzian functions.

**Figure 8 nanomaterials-11-01035-f008:**
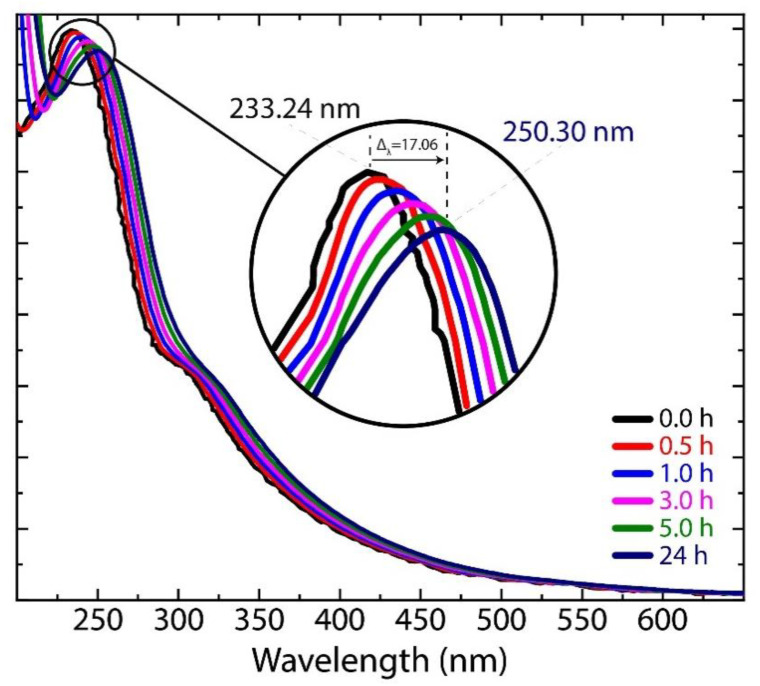
UV-vis spectra of GO subject to different drying times.

**Figure 9 nanomaterials-11-01035-f009:**
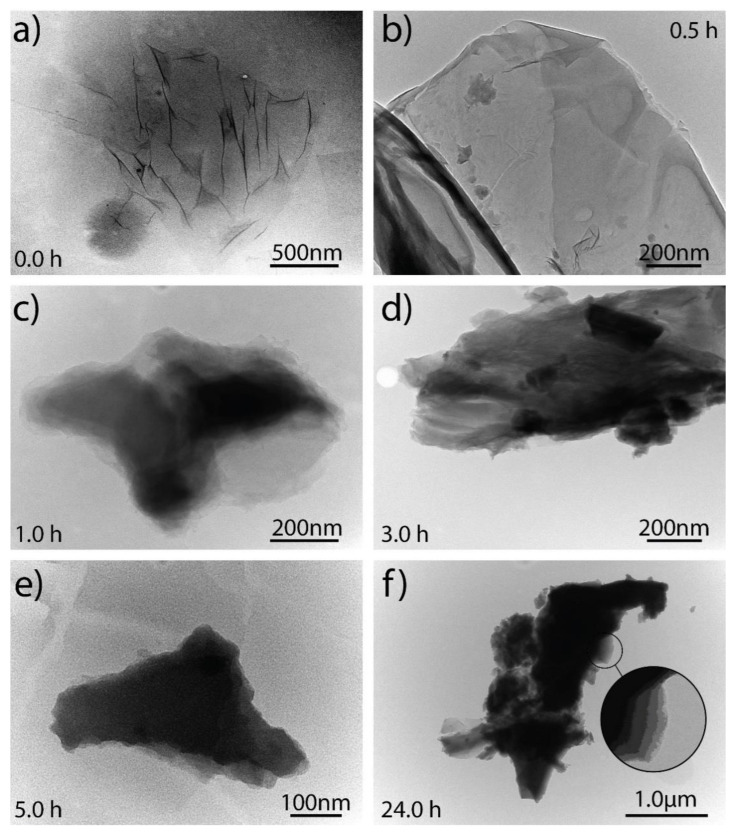
TEM analyses of GO subject to 80 °C and considering different drying times: (**a**) 0.0 h, (**b**) 0.5 h, (**c**) 1.0 h, (**d**) 3.0 h, (**e**) 5.0 h, and (**f**) 24 h.

**Table 1 nanomaterials-11-01035-t001:** Peak position and full-width at half maximum (FWHM) of GO at different drying times in the region from 1000 to 2000 cm^−1^. The FWHM was obtained using Lorentzian fitting.

	D*-FWHM (cm^−1^)	D-FWHM (cm^−1^)	D**-FWHM (cm^−1^)	G-FWHM (cm^−1^)	D’-FWHM (cm^−1^)
0.0 h	–	1348–122	1511–122	1576–58	1608–36
0.5 h	1121–61	1347–127	1498–131	1571–64	1606–39
1.0 h	1135–64	1346–126	1514–126	1576–60	1608–37
3.0 h	1135–46	1347–147	1511–129	1572–65	1605–41
5.0 h	1125–174	1346–151	1517–127	1576–63	1606–41
24 h	1124–89	1344–143	1518–130	1577–62	1607–40

**Table 2 nanomaterials-11-01035-t002:** Peak position and full-width at half maximum (FWHM) of GO at different drying times in the region from 2300 to 3000 cm^−1^. The FWHM was obtained using Lorentzian fitting.

	2D_1A_-FWHM (cm^−1^)	2D_2A_-FWHM (cm^−1^)	D+G-FWHM (cm^−1^)
0.0 h	2679–246	2755–140	2937–207
0.5 h	2685–259	2751–46	2930–222
1.0 h	2685–259	2751–46	2930–222
3.0 h	2669–255	2755–141	2928–251
5.0 h	2653–200	2750–200	2921–144
24 h	2653–188	2749–175	2922–131

## Data Availability

The data that supports the findings of this study are available within the article. Any additional data relevant to this study are available from the author upon reasonable request.

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
