# Peer review of "Drying-Time Study in Graphene Oxide"

_nanomaterials, 2021, doi:10.3390/nano11041035_

Round 1

Reviewer 1 Report

This manuscript describes a study of the effect of drying on the spectral characteristics of graphene oxide. The work is interesting and in general is well presented and discussed so I recommend acceptance after revision.

Comments:

- From a practical point of view, it would be interesting to study the reverse process, i.e rehydration of the dried GO, and how does it affects the morphology of the material. This can be simply done by recording TEM images before and after rehydration, ideally of the same sample. I strongly suggest the authors to do this experiment as it would enhance the presentation of the results and the quality of the paper.

Author Response

Response to Reviewer 1 Comments

We thank the referee for his/her very valuable comments. We also thank the referee for considering that our work is interesting and well presented to be considered for publication in Nanomaterials - MDPI”

We have considered your suggestion, as follow:

Point 1: From a practical point of view, it would be interesting to study the reverse process, i.e rehydration of the dried GO, and how does it affects the morphology of the material. This can be simply done by recording TEM images before and after rehydration, ideally of the same sample. I strongly suggest the authors to do this experiment as it would enhance the presentation of the results and the quality of the paper.

Response 1: Thank you very much for this interesting suggestion. We would love to meet this requirement based on TEM measurements. However, please, consider the following points:

  1. All the experimentation was carried out at the Yachay Tech University located in Ecuador and TEM measurements can be done at the University of Calabria located in Italy.
  2. Given the current conditions due to the Pandemic, it is impossible to make a detailed study of the effect of rehydration, and to prepare samples for shipment to Europe from South America.
  3. Keeping the point 3, we are also limited to work at the Laboratories for long period of time.
  4. The requested study is limited by the short time given by the Editor (7 days)

To meet this requirement, we have carried out UV visible and Raman measurements on re-hydrated samples after they have been dried at 1 h, 5 h, and 24 h. The resulting Figures are reported in a Supplementary Material and the corresponding discussion in the revised manuscript (highlighted in red).

The main findings are:

  • The UV-vis spectra show a blueshift compared to the same samples subject to drying time experiment.
  • The D* band is absent and the 2D2A band is barely perceptible in the Raman spectra.

These results open up new questions, for this reason, we really appreciate this suggestion, which needs to be considered in a more extended study, which cannot be addressed in our communication at this time.

Reviewer 2 Report

In this work,  the effect of drying time (from 0.0 h to 24 h) on eco- 
friendly-prepared graphene oxide was studied by Raman spectroscopy. The Raman spectrum of graphite and few-layer graphene were also discussed.  The manuscript is well written but the scientific work should have a  more general overlap. n the work it is necessary to describe more precisely the relationship between the properties of the investigated materials and the methodology of their preparation, resp. how by a  modification of the technology can achieve a relevant result and how to analyze the states and properties of the material. Also, results published for relevant polycrystalline diamond materials (Raman spectroscopy) should be discussed - I. Kratochvílová, et al:  Nanosized polycrystalline diamond cladding for surface protection of zirconium nuclear fuel tubes,  Journal of Materials Processing Technology 214 (2014) 2600-2605. 

Author Response

Response to Reviewer 2 Comments

We thank the referee for his/her very valuable observations, which have indeed helped us improving the present communication, and also for considering that our work “is well written…” We have carefully scrutinized the referee comment and suggested reference:

Point 1: The manuscript is well written but the scientific work should have a  more general overlap. n the work it is necessary to describe more precisely the relationship between the properties of the investigated materials and the methodology of their preparation. resp. how by a  modification of the technology can achieve a relevant result and how to analyze the states and properties of the material.

Response 1: Thank the Reviewer for this important comment. We have restructured the results and discussions section, particularly what is related to the GO experiment subjected to the drying time, giving an overview and cancelling some unnecessary paragraphs. We have also focused on how the process could be controlled or optimized based on the reported findings.
Please, see the revised manuscript (highlighted in red)

Point 2: Also, results published for relevant polycrystalline diamond materials (Raman spectroscopy) should be discussed - I. Kratochvílová, et al:  Nanosized polycrystalline diamond cladding for surface protection of zirconium nuclear fuel tubes,  Journal of Materials Processing Technology 214 (2014) 2600-2605.

Response 2: At the time the manuscript was summited we were unaware of this article. Now we have added it as Ref. [31] in the Introduction section and discussed, particularly, in the Raman spectrum of non-dried Graphene oxide. Please, see the revised manuscript (highlighted in red)

Reviewer 3 Report

The manuscript by Tene and co-workers investigated the drying time of graphene samples. The work has some merits, and although the effect of drying has already been studied, the authors present new insights. However, there are several minor and major points to be address prior to further consideration.

1) The eco-friendly buzz word should be removed from the title and other parts of the manuscript. The presented work does not contribute to the development of an eco-friendly process but investigates the drying time. The so-called eco-friendly process was developed under references 29 and 37 already.

2) The effect of drying on graphene quality was already investigated from different angles, and these should be mentioned (10.4028/www.scientific.net/AMR.554-556.597).

3) The authors should comments on the reproducibility of their work. Errors are not presented and discussed, which needs to be corrected.

4) Some critical comments should be added in the manuscript; what are the main drawbacks and limitations of the study? How can the results be practically implemented by the community, and what are the authors suggestions?

5) In line 38, environmental remediation as emerging application should be mentioned (10.1016/j.apmt.2020.100878).

6) What is the dependence of the results on the quality of the graphite used to prepare the graphene, or the method used to obtain the graphene from the graphite?

7) In line 28-19, clarify how exactly the presented research can ‘immediately help’ the production. The authors repeat this in the conclusion as well. It is unclear how the results are of immediate help. The authors should elaborate on the potential impact of the work, and the usefulness of the results.

8) The results section has too many references, and therefore it is difficult to identify what the new results are. It seems that everything has been reported before. The authors should modify the results section accordingly, and ensure that the novel findings come across as such.

Author Response

Response to Reviewer 3 Comments

We thank the referee for a careful reading of the manuscript and for his/her very valuable comments, which have indeed helped us improving the manuscript. We also thank the referee for considering that our work presents “new insights”, which can be considered eligible to be published considering comments and suggestions.

We have addressed all the points, as follow:

Point 1. The eco-friendly buzz word should be removed from the title and other parts of the manuscript. The presented work does not contribute to the development of an eco-friendly process but investigates the drying time. The so-called eco-friendly process was developed under references 29 and 37 already.

Response 1. Thank you the Reviewer for this useful comment. We have removed from the title and main text the eco-friendly term. We were trying to emphasize the investigation on this type of samples never reported. We accept this suggestion, hoping readers will take our approach implicitly.

Point 2. The effect of drying on graphene quality was already investigated from different angles, and these should be mentioned (10.4028/www.scientific.net/AMR.554-556.597).

Response 2. At the time our work was submitted, we were not aware of this research. Thank you, we have added it as Ref [39] and mentioned it at the beginning of the Results and Discussions section.

Point 3. The authors should comments on the reproducibility of their work. Errors are not presented and discussed, which needs to be corrected.

Response 3. Thank you for this very important observation. As mentioned, this communication follows a series of works from the production and use of our material in the removal of colorants and heavy metals. Several replicas of our method have been made. The process has been detailed and validated with more spectrocopical and morphological techniques, please, see References [25] and [37]. Additionally, we have added the propagation uncertainty in Figures 5 and 7.

Point 4. Some critical comments should be added in the manuscript; what are the main drawbacks and limitations of the study? How can the results be practically implemented by the community, and what are the authors suggestions?

Response 4. Thank you the Reviewer for these interesting questions. To address this point without losing the main objective of our work, we have expanded the conclusions section (highlighted in red).

Point 5. In line 38, environmental remediation as emerging application should be mentioned (10.1016/j.apmt.2020.100878).

Response 5. Thank you for the suggested work, an interesting application. We have added it as Ref. [26].

Point 6. What is the dependence of the results on the quality of the graphite used to prepare the graphene, or the method used to obtain the graphene from the graphite?

Response 6. Thanks for this observation. At this time, we cannot argue on this fact because we have always used graphite powder to prepare graphene oxide following synthesis and applications. We would like to have results to properly discuss the effect of the type of graphite, however, it is not the objective of the present communication, and it deserves a more extended work. In fact, large lateral graphite or expanded graphite could present different results compared to those reported here. Instead, we have suggested that future work should consider this point. The latter is added in the conclusions (highlighted in red).

Point 7. In line 28-19, clarify how exactly the presented research can ‘immediately help’ the production. The authors repeat this in the conclusion as well. It is unclear how the results are of immediate help. The authors should elaborate on the potential impact of the work, and the usefulness of the results.

Response 7. Although our results do not directly help a large-scale production of GO, they can be used as a guideline for the control of the technical parameters involved in the oxidation and reduction processes, where dried oxidized graphenes are needed. But we accept the suggestion. This statement has been removed from the Abstract and rewritten in the conclusions (highlighted in red).

Point 8. The results section has too many references, and therefore it is difficult to identify what the new results are. It seems that everything has been reported before. The authors should modify the results section accordingly, and ensure that the novel findings come across as such.

Response 8. Thank you for this important comment. To improve the results and discussions section, we have canceled some redundant paragraphs and have removed unnecessary references so that the reader clearly notes the new results reported.

Round 2

Reviewer 1 Report

I understand the difficulties mentioned by the authors in recording TEM and appreciate the alternative experiments carried out. The manuscript can be accepted in its current form.

Reviewer 2 Report

To be published.

Reviewer 3 Report

The authors have addressed some of the comments and the manuscript improved somewhat.